# Mesenchymal Stem Cells for Regenerative Medicine

**DOI:** 10.3390/cells8080886

**Published:** 2019-08-13

**Authors:** Yu Han, Xuezhou Li, Yanbo Zhang, Yuping Han, Fei Chang, Jianxun Ding

**Affiliations:** 1Department of Orthopedics, The Second Hospital of Jilin University, 218 Ziqiang Street, Changchun 130041, China; 2Key Laboratory of Polymer Ecomaterials, Changchun Institute of Applied Chemistry, Chinese Academy of Sciences, 5625 Renmin Street, Changchun 130022, China; 3Department of Orthopedics, China-Japan Union Hospital of Jilin University, 126 Xiantai Street, Changchun 130033, China; 4Department of Urology, China-Japan Union Hospital of Jilin University, 126 Xiantai Street, Changchun 130033, China

**Keywords:** mesenchymal stem cell, extraction, cell differentiation, biomedical application

## Abstract

In recent decades, the biomedical applications of mesenchymal stem cells (MSCs) have attracted increasing attention. MSCs are easily extracted from the bone marrow, fat, and synovium, and differentiate into various cell lineages according to the requirements of specific biomedical applications. As MSCs do not express significant histocompatibility complexes and immune stimulating molecules, they are not detected by immune surveillance and do not lead to graft rejection after transplantation. These properties make them competent biomedical candidates, especially in tissue engineering. We present a brief overview of MSC extraction methods and subsequent potential for differentiation, and a comprehensive overview of their preclinical and clinical applications in regenerative medicine, and discuss future challenges.

## 1. Introduction

Since the discovery of spindle-shaped, bone marrow-derived plastic-adherent cells in the mid-1970s [1], science has come a long way, and studies have found that these cells could differentiate into osteoblasts and chondrocytes [2,3]. Techniques for extraction, culture, and induction of mesenchymal stem cells (MSCs) have improved, with almost all MSC types derived from various tissues now capable of differentiation into osteocytes and end-stage lineages [4]. The rapid development of molecular biology and transplantation techniques has benefitted MSC applications in regenerative medicine.

MSCs are an ideal cell source for tissue regeneration, owing to the excellent properties as follows. MSCs exist in almost all tissues, including bone marrow, adipose, and synovium [5], and are easily extracted. MSCs can differentiate into almost any end-stage lineage cells to enable their seeding in specific scaffolds (Figure 1) [6]. Their immunological properties, including anti-inflammatory, immunoregulatory, and immunosuppressive capacities, contribute to their potential role as immune tolerant agents [7,8].

Numerous studies have explored MSCs for tissue regeneration in several animal models in vitro; trials have not been limited to preclinical validation. Several clinical reports verify the potential efficacy of MSC-based cell therapy; although its effectiveness remains limited, the outcomes are inspiring. We present a brief overview of MSC extraction methods and subsequent potential for differentiation and provide a comprehensive overview of future applications of various MSCs in regenerative medicine, as well as the challenges.

## 2. Discovery and Extraction of MSCs from Different Sources

The rich source of MSCs is the critical basis for their extensive researches and applications. It is known that MSCs can be isolated from various tissues, such as bone marrow, adipose, and synovium, and human umbilical cord blood, and bone marrow is one of the essential sources of MSCs.

MSCs exist in various tissues and organs apart from bone marrow, with multilineage cells from human umbilical cord blood, first reported in early 2000 [9]. Adipose tissue was subsequently demonstrated as a rich source of MSCs in 2001 [10], and synovium-derived MSCs (SMSCs) were successfully isolated [11]. MSCs from other tissues or organs were detected, and protocols were established for their extraction, identification, and culture (Figure 2 and Table 1) [12,13,14,15,16,17,18,19,20,21,22,23,24,25,26,27,28,29,30]. Figure 2 and Table 1 describe the general protocols used for MSC extraction. Briefly, the process involves isolation of various tissues, digestion to obtain cells, and culture for three to five days, followed by discarding non-adherent cells and continuous culture of adherent cells to the desired passage. The primary culture medium for MSCs includes low-glucose Dulbecco’s modified Eagle medium (LG-DMEM) with 1% (*W*/*V*) antibiotic/antimycotic and 10% (*V*/*V*) fetal bovine serum (FBS). Additionally, Table 1 lists a variety of markers expressed on the MSC surface. Notably, rabbit is the most frequently used animal model for experiments, involving cartilage or bone tissue regeneration, and should receive increased focus concerning MSC identification. Moreover, the surface markers of rabbit tissue-derived MSCs require further verification.

## 3. Differentiation Potentials of MSC Types

The multi-directional differentiation potential is one of the most critical characteristics of MSCs. In addition, different tissue sources affect the differentiation tendency and proliferation capability of MSCs.

There is an increasing number of publications addressing the heterogeneity of MSCs [47]. The transcriptome, proteome, immunophenotype, and immunomodulatory activities of various MSC types differ, implying that MSCs exhibit unique differentiation potentials. As a critical MSC-specific property, differentiation potential affects MSC fate; different tissue-derived MSCs display distinct tendencies to differentiate into different end-stage lineage cells, such as osteoblasts and chondrocytes. As a critical source of MSCs for tissue engineering, bone marrow-derived MSCs (BMSCs) exhibit superior capacities for osteogenesis and chondrogenesis under standard differentiation protocols [48], and SMSCs show more significant proliferation and chondrogenic potential than adipose-derived MSCs (ADSCs) [49]. Umbilical cord blood-derived MSCs (UCB-MSCs) exhibit biological advantages relative to other adult sources, including their capability for longer culture times, larger-scale expansion, more significant retardation of senescence, and higher anti-inflammatory effects [50]. Researchers must choose the desired MSC type according to the specific purpose. Table 2 summarizes fundamental and in vivo experiments and identifies differentiation conditions based on previous studies.

## 4. MSC-Based Regenerative Medicine

So far, MSCs have been widely studied and applied in regenerative medicine. In this section, we summarize reports concerning the latest preclinical and clinical trials of various MSC types for tissue engineering. The topics mainly focus on the reconstruction of fragile tissues, including those associated with the musculoskeletal system, nervous system, myocardium, liver, cornea, trachea, and skin, as shown in Figure 3.

### 4.1. Bone Regeneration

Bone defects frequently accompany recovery from trauma, revision arthroplasty, or tumor resection surgeries. Autologous bone grafting represents the gold standard therapeutic strategy, despite its many drawbacks, including (1) the limited supply of autologous bone, (2) increased operation time and blood loss, (3) temporary disruption of bone structure in the donor site, and (4) donor site morbidity [51]. Allografting carries a risk of disease and/or infection [52]. Therefore, MSC-based bone regeneration is considered an optimal approach [53].

The MSC osteoblast-differentiation capacity has been identified [2,3], with BMSCs representing the most frequently applied cells for osteoblast differentiation [2]. Comparative studies evaluating the osteogenic ability of other MSC types yielded no definitive conclusions. By contrast, UCB-MSCs show better angiogenic capacity, supporting more abundant blood supply during bone regeneration [54], which promotes rapid tissue reconstruction. In addition to BMSCs, human dental pulp stem cells (hDPSCs) show excellent vascular differentiation potential while differentiating into osteoblasts, which subsequently support bone regeneration [55]. However, these hDPSCs are screened from a stromal vascular dental pulp fraction; therefore, this represents a limited source for further research and application. Since ADSCs can be routinely isolated from lipoaspirate with a high degree of purity with minimal donor site morbidity or patient discomfort, ADSCs are considered to have the most significant potential as a primary source for clinical bone tissue engineering [48,56,57]. Additional comparative and screening studies are necessary to identify other cell sources with applications in bone reconstruction.

Stimulating factors play an important role in directing MSC differentiation into target cells in vitro. The most commonly used inducing factor for osteogenesis is the bone morphogenetic protein-2 (BMP-2), which is usually immobilized on scaffolds to promote osteoblast differentiation. BMP-2 exhibits a strong osteogenic ability, which can be tested by the osteoblast activity and/or expression of bone markers, such as alkaline phosphatase (ALP), osteopontin (OPN), and osteocalcin (OCN) [58,59,60]. BMP-7 activates the transforming growth factor-β (TGF-β) /SMAD signaling in CD105^+^ MSCs to enhance the expression of osteogenesis-related genes [61]; Wnt11 enhances the osteogenic potential of BMP-9 [62]. Nano-hydroxyapatite [58] and strontium [63] are used as osteogenic regulators in tissue engineering to promote osteogenic differentiation of MSCs while changing the physical properties of the scaffolds.

Studies of MSC-based cell therapy for bone defects and the use of novel scaffolds describe inspiring advances in vitro and in vivo [64,65]. Clinical applications of MSCs in bone reconstruction have been described, including those involving implantation of scaffolds seeded with MSCs into bone defect sites. Specifically, dentists have used this technique to address alveolar cleft defects, jaw defect reconstruction, and maxillary sinus augmentation, with excellent outcomes [66,67,68]. Defects in or non-union of human tubular bone have been tentatively treated via local implantation of MSCs with or without scaffolds [69,70].

### 4.2. Cartilage Repair

Cartilage defect repair is one of the significant challenges faced by orthopedic surgeons. Due to the inherent avascular nature of cartilage and the proliferation of mature chondrocytes, cartilage is greatly limited in its ability to repair itself. Currently, the clinically applied cartilage repair techniques, such as bone marrow stimulation and osteochondral transplantation, have their limitations. Fibrocartilage produced by bone marrow stimulation is not strong enough, and grafts for osteochondral transplantation are challenging to integrate.

MSCs offer a new strategy for the repair of damaged cartilage, as they can differentiate into chondrocytes [2,3]. An integrated cartilage reconstruction unit comprises cells, scaffolds, and stimulatory factors, with BMSCs [71], ADSCs [56], and SMSCs [72], used as the primary cell sources. Among these, BMSCs displayed better chondrogenic capacity in vitro and in vivo [33,34,71], although SMSCs show better proliferation and differentiation potential and less hypertrophy than BMSCs and ADSCs [72]. Cartilage reconstruction requires a combination of multiple stimulating factors; co-culture of chondrocytes and MSCs would achieve better results than the application of MSCs alone [73].

Novel bioactive three-dimensional (3D) scaffolds, such as hydrogels [55] and electrospun scaffolds [74], have undergone constant improvement. These scaffolds provide an optimal 3D microenvironment for cartilage regeneration. Moreover, hydrogels can regulate MSC proliferation and differentiation due to their high water content and biocompatibility and similar properties to the extracellular matrix (ECM) [55]. Injectable hydrogels enable minimally invasive treatment of large areas of cartilage defects [75]; thus, hydrogels loaded with MSCs and stimulating factors are highly efficacious for the repair of cartilage damage. The development of electrospun scaffolds suggests that the arrangement of nanofibers also affects cell differentiation and provides a different approach to cartilage repair [74]. Stimulating factors are necessary for cartilage engineering and responsible for inducing, accelerating, and/or enhancing cartilage formation. Common stimulating factors include BMP-2/-4, insulin-like growth factor (IGF)-1, and TGF-β1/-β3 [76,77,78]. Moreover, physical stimuli, such as hydrostatic pressure and dynamic compression, have been explored to induce MSC-mediated cartilage formation [79].

The first preclinical trial of MSC application for cartilage repair occurred in 1994 [80]. MSCs were seeded into a collagen (Col) gel to treat a full-thickness defect in rabbit femoral cartilage, resulting in better outcomes than those observed in a control group. This defect model was subsequently used as a classical cartilage defect model for cartilage regeneration; subsequently, numerous trials have been conducted in both animal models and humans to evaluate MSC-based therapy for cartilage damage [81].

Despite clinical trials being conducted, there are no commercially available products for MSC-based cartilage reconstruction [82,83]. Several studies investigated the effects of expanded MSCs in vivo on damage to human articular cartilage. Transplantation of expanded autologous BMSCs improved cartilage quality in patients with chronic knee osteoarthritis [83], although the clinical improvement was not significant [82]. Other studies reported the injection of allogeneic MSCs into joints in the presence or absence of pre-mixing with autologous chondrocytes [81]. All the clinical outcomes indicated the safety of these therapeutic approaches, and their ability to relieve some symptoms, although their ability to repair the effects of cartilage damage was not always apparent. MSC transplantation showed better results for early lesions [81].

There remain many challenges for MSC-based cartilage regeneration, including the identification of optimal cell sources. Additional studies are needed to enable the use of MSC-based materials as commercial products for implantation to promote cartilage regeneration.

### 4.3. Regeneration of Other Musculoskeletal Tissues

Recent studies investigated the MSC-mediated regeneration of musculoskeletal tissues outside the bone and cartilage, including the meniscus, tendons and ligaments, and intervertebral discs (IVD).

Meniscus regeneration has received increasing attention. Intra-articular administration of MSCs to promote meniscal regeneration was first performed with favorable outcomes [84]. Similar to its use for cartilage regeneration, hydrogels [85] and electrospun scaffolds [86] loaded with MSCs were used to reconstruct the meniscus. Moreover, the meniscus-derived decellularized matrix shows better histocompatibility and is more capable of inducing MSC differentiation as compared with natural or synthetic polymer materials [87]. Scaffold-free tissue-engineered constructs show promise as an MSC-based implantation technique to repair meniscal lesions [88]. Tarafder et al. [89] proposed the recruitment of synovial MSCs through connective tissue TGF and TGF-β3 to repair meniscus injury, thereby avoiding the disadvantages of cell-based techniques. Mechanical stimulation is crucial for meniscus growth and maintenance, with mechanical stimuli, such as dynamic compression and tensile loads applied for meniscus repair [90]. Although satisfactory results were obtained in animal models, there remains a lack of evidence in humans regarding the capability of MSCs for forming durable tissues similar to the meniscus [91].

Tendon injury is a common problem associated with sports [92]. BMP-14 induces myogenic differentiation of BMSCs via the sirtuin-1−Janus *N*-terminal kinase (JNK)/SMAD1-peroxisome proliferator-activated receptor-γ signaling pathway [93]. Studies describing tendogenic differentiation of MSCs were not limited to stimulating factors [94] and scaffolds [95] but also referred to mechanical stimuli that play essential roles in MSC differentiation into tendon lineages. Uniaxial cyclic stretching promoted tendogenic differentiation of MSCs in vitro and in vivo [96]; however, MSCs did not repair tendon injury but only delayed lesion progression [97].

With the increasing age of populations, IVD degeneration has become prevalent. MSCs represent promising candidates for disc regeneration; scaffolds made of Col provide readily available support for chondrogenic differentiation of MSCs in vitro, although the phenotype of the differentiated MSCs is not yet equivalent to that of nucleus pulposus (NP) cells [98]. The acellular matrices derived from NP cells stimulated by TGF-β3 also enhance MSC differentiation [99]. Transfection of adenoviral expression of SOX-9 and BMP-2 in BMSCs increased Col II and aggrecan expression, and promoted IVD repair [100]. Varma et al. [101] used a hydrogel loaded with two different concentrations of MSCs to repair NP, and showed that MSC inoculation at a lower density resulted in a better NP-specific matrix phenotype. A systematic review of MSC-based cell therapy for IVD indicated the safety and effectiveness of short-term MSC transplantation, as well as the necessity for human-based clinical trials [102]. In 2011, expanded autologous BMSCs injected into patients with lumbar disc degeneration revealed several advantages and better prognosis relative to the current gold standard treatments [103]. Clinical percutaneous injection of autologous bone marrow concentrate cells into a patient with degenerative IVD resulted in decreased lumbar discogenic pain within 13 years [104]. Clinical studies [105] indicated that MSC transplantation represents a safe treatment option for degenerative IVD; the specific effects need verification by additional clinical trials.

### 4.4. Central Nervous System Rebuilding

The adult central nervous system (CNS) lacks the ability to repair damaged neurons, so the damage of CNS is irreversible, and there is currently no effective repair method for CNS injury in clinical practice repair. In the area of CNS regeneration, MSC-based therapy mainly focuses on two areas: Damage or injury of the CNS caused by severe trauma and continuous ischemia and CNS dysfunction caused by neurologic diseases. To date, BMSCs and ADSCs are the most extensively studied cell sources for CNS repair, with each showing similar neuronal differentiation potential [35,36]. BMSCs can reduce scar formation around spinal cord injury (SCI) lesions and promoted axonal regeneration [106]; however, ADSCs might represent a more suitable cell source owing to their easy extraction and abundant sources. ADSCs inhibit inflammation of the nervous system and improve the recovery of function from traumatic brain injury via neural stem cells [107]. UCB-MSCs can be induced to differentiate into neuron-like cells in vitro [37]; DPSCs can differentiate into neurons and express multiple factors that promote neuronal and axonal regeneration [108].

MSC expression of neuronal or astrocytic markers has been observed in vitro [109] and in vivo [110]. To promote MSC-based CNS restoration, gene-modified MSCs, such as neurotrophin-3-transferred BMSCs, showed improved neuronal differentiation in vivo [111]. Persistent release of specific cytokines and growth factors, which can facilitate neurogenesis, angiogenesis, and synaptogenesis, creates a favorable microenvironment for angiogenesis or remyelination during reconstruction [112]. IGF-1-transfected spinal cord-derived neural stem cells displayed higher viability and the ability to differentiate into oligodendrocytes [113]. Moreover, MSCs can induce T cell tolerance and release of paracrine anti-inflammatory factors, such as TGF-β, that promote neuroprotective effects [114].

Animal models of traumatic and ischemic brain injury or SCI have been used to evaluate MSC-based therapy [115,116]. A meta-analysis of 1568 rats with traumatic SCI showed that MSC therapy provided a substantial beneficial effect on locomotor recovery [116]. Clinical studies indicated MSC-based therapy as a safe and feasible technique for patients with SCI and/or traumatic brain injury [117,118]. Migration of MSC pretreated under hypoxic conditions to the peri-cerebral injury area of cerebral hemorrhagic stroke victimized rats resulted in the release of various growth factors to promote neurogenesis and neurological recovery [119]. For neurological diseases, non-expanded or expanded MSCs have been widely used for the treatment of multiple sclerosis [120], amyotrophic lateral sclerosis [121], ischemic stroke [122], and Parkinson’s disease [123]. Most of the beneficial effects of MSCs on neurological diseases are associated with their immunomodulatory and neuroprotective properties exerted following local injection of non-expanded or expanded autologous MSCs, with clinical trials assessing their ability to achieve promising outcomes and different degrees of remission.

### 4.5. Peripheral Nervous System Rebuilding

Peripheral nervous system (PNS) injury is mainly caused by severe trauma, usually accompanied by bone fracture and vascular damage. Autologous nerve grafting (autologous nerve bridging) is the gold standard for peripheral nerve repair; however, limited donor nerve resources and other issues preclude the search for new therapeutic strategies. Schwann cells, neurotrophic factors, and anti-inflammatory cells work together to promote peripheral nerve regeneration, with this process involving axonal sprouting and fiber myelination.

There are few comparative studies concerning the effects of different MSC types on animal models of peripheral nerve injury. ADSCs are more suitable cell sources for neural regeneration in vitro [35], and a nerve growth factor transcript has been identified in ADSC-secreted nanovesicles that promotes synaptic growth in vitro and repair of sciatic nerve damage in vivo [124]. Sun et al. [125] proposed a new protocol called intermittent induction that alternates complete and incomplete induction media to induce ADSC differentiation into SLCs. Compared with traditional protocols, SLCs obtained by intermittent induction secrete neurotrophic factors and promote axonal growth in vitro and more effectively repair rat sciatic nerve injury in vivo. Notably, SLCs seeded in acellular nerve grafts show better functional recovery as compared with MSCs [126]. BMSCs [111], ADSCs [127], and UCB-MSCs [128] have also been seeded onto a variety of biodegradable scaffolds, with almost all resulting in better recovery relative to controls. MSCs form a neuroblast-like sheath following transplantation at the site of nerve injury and secrete neurotrophic factors that provide physical and chemical barriers for the inner nerve fibers [110]. Compared with polymers, acellular neural matrix hydrogels show better biocompatibility and tissue specificity and support Schwann cell proliferation in vitro and repair rat sciatic nerve defects in vivo [129]. Furthermore, 3D-bioprinting technology has enabled the development of 3D scaffolds with complex structures to address the challenges of nerve tissue regeneration [130].

For nerve regeneration studies, sciatic nerve crush and nerve gap animal models were established [131,132], and the effects of local implantation [133] or intravenous administration [134,135] of neural stem cells or MSCs in peripheral nerve injury models have been investigated, resulting in excellent outcomes relative to controls. ADSCs displayed relevant therapeutic potential not only via their direct release of growth factors but also through the indirect modulation of neurocyte behavior in an animal model of acute axonal injury [134]. Moreover, intravenously infused MSCs ameliorated function recovery post-acute peripheral nerve injury in a sciatic nerve crush model [135].

Although preclinical studies show the feasibility of MSC-based therapy in animal models of peripheral nerve injury, there are few reports of its clinical application [126].

### 4.6. Myocardium Restoration

Cardiac disease is characterized by substantial morbidity and mortality, and serious adverse consequences. In addition to congenital heart disease, almost all cardiac diseases involve insufficient blood supply to critical regions, resulting in myocardial damage and necrosis. Although myocardium has limited regenerative capacity, restoration of severe damage to cardiomyocytes due to catastrophic myocardial infarction or other myocardial diseases is inadequate.

A role for MSC in attenuating myocardium damage was first reported in 2002 [136]; purified BMSCs engrafted in the murine myocardium appeared to differentiate into cardiomyocytes. Several subsequent studies evaluated the potential of different MSC sources to differentiate into cardiomyocytes [38,39,40], finding ADSCs as the most suitable. Spraying was found to be a more cost-effective and less invasive method for transferring ADSCs into a pig heart infarction model to promote cardiac function recovery [137]. MSC-specific mechanisms associated with the repair of damaged cardiomyocytes involve three factors: (1) myogenic and angiogenic capacities; (2) the ability to supply massive amounts of angiogenic, anti-apoptotic, and mitogenic factors; and (3) the inhibition of myocardial fibrosis (Figure 4) [138,139]. Butler et al. [140] demonstrated the safety of MSC therapy, and that it improved the left ventricular ejection fraction in patients with non-ischemic cardiomyopathy via its immunomodulatory effects. Co-culture of MSCs with cardiomyocytes promotes resistance to high oxidative stress in heart tissue after myocardial infarction [141]. 5-Azacytidine is an effective factor for inducing MSC differentiation into cardiomyocytes [142]. IGF-1-transfected MSCs protected the myocardium from fibrosis and cardiomyocyte apoptosis and reduced infarct size after myocardial infarction in rats [143]. Interleukin (IL)-7 enhances the fusion of MSCs with cardiomyocytes to improve cardiac function, with this attributed to the ability of IL-7 to promote cell proliferation and support damaged myocardial regeneration [144]. In addition to their ability to differentiate into cardiomyocytes, MSCs promoted angiogenesis by secreting vascular endothelial growth factor (VEGF) in a critical limb ischemia model [145], resulting in cardiac reconstruction.

MSC-based therapy is a feasible strategy to improve cardiac function, according to preclinical and clinical findings [146]. Furthermore, MSC therapy has been intensively investigated as a treatment for myocardial infarction [147], peripheral ischemic vascular diseases [148], dilated cardiomyopathy [138], and pulmonary hypertension [149].

### 4.7. Liver Regeneration

The liver is an essential human organ, failure of which causes fatal illnesses. Until recently, the only effective therapy for liver failure was organ transplantation [150]; however, the availability of transplantable livers is scarce, and many patients do not survive the wait time for transplantation. Cell therapy provides a possible solution by either building a partial or full liver for transplantation or addressing the damage to the liver. Although hepatocytes isolated from the livers of donors have been studied, their therapeutic efficacy is questionable owing to their immunogenicity. Recently, MSCs have become a therapeutic option owing to their ability to exert potent immunosuppressive and anti-inflammatory effects [8]. Importantly, the multilineage potential of MSCs to differentiate into different types of end-stage cells, including hepatocyte, makes them an attractive candidate for liver-specific therapeutics [151].

The clinical use of MSCs to treat liver failure is mainly based on their immunomodulatory capacity [152]. Preclinical and clinical studies revealed the mechanisms associated with immunoregulation following MSC-based treatment, including transdifferentiation, fusion, inhibition of Col deposition, paracrine effects, modulation of matrix metalloproteinase expression and activity, neoangiogenesis, and vascular support [153,154,155]. A clinical phase II trial showed that transplantation of autologous BMSCs inhibited tissue fibrosis and improved liver function in alcoholic cirrhosis [156]; intravenous injection of allogeneic MSCs positively affected the treatment of autoimmune cirrhosis [157]. Administration of UCB-MSCs reduces the liver inflammatory response and hepatocyte injury, as well as the possibility of liver failure, by inhibiting T cell and B cell proliferation and upregulating the levels of regulatory T cells [158].

In addition to their immunomodulatory effects, MSCs can differentiate into hepatocytes to promote liver regeneration. MSCs can differentiate into several types of liver cells under certain physiopathological conditions [151]. Functional hepatocyte-like cells were obtained from multipotent adult progenitor cells [159]. Hepatocyte growth factor (HGF) and oncostatin M have been used to successfully induce human BMSCs and human UCB-MSCs to differentiate into hepatocyte-like cells [160]. Hepatocyte-like cells have been derived from BMSCs [161], ADSCs [41,42], UCB-MSCs [43], and placenta-derived MSCs (PDSCs) [44] via appropriate culture conditions, including co-culture with hepatocytes on 2D or 3D scaffolds [142], floating culture, or treatment with serum collected from rats after partial hepatectomy [162]. Liver reconstruction using differentiated MSCs has only been demonstrated in the animal models, although recently, a phase II trial showed that transplantation of differentiated autologous MSCs could be used as a potential clinical treatment for liver cirrhosis [14].

There remain two obstacles to MSC-based liver regeneration: transplanted MSCs cannot efficiently differentiate into hepatocyte-like cells, and transplanted MSCs might promote fibrogenesis [163]. Moreover, a previous study suggested that transplanted MSCs might contribute to the myofibroblast pool to enhance fibrotic processes within the liver [164]. However, a clinical study [165] reported that MSC injections were sufficiently safe, but their immunosuppressive effects following liver transplantation were insufficient to replace immunosuppressants, despite the severe side effects of immunosuppressants commonly used in clinical practice.

### 4.8. Corneal Reconstruction

The cornea represents a transparent avascular connective tissue that provides most of the refractive ability of the eye and acts as the primary barrier against infection and mechanical damage to internal structures. Because the cornea is fragile and directly exposed to the external environment, a variety of clinical disorders, such as aniridia and Stevens-Johnson syndrome, or chemical, mechanical, and thermal injury potentially causes corneal injury. Human corneal epithelial cells can be renewed by stem cells located in the peripheral region of the cornea, containing limbal epithelial stem cells (LESCs). However, if the entire corneal layer is damaged, vascularization, conjunctivalization, keratinization, corneal scarring, and opacification occur and lead to impaired vision and even blindness. Currently, keratoplasty is the primary method used for corneal restoration, because sources of donor tissue are restricted, and post-operation immune rejection compromises transplantation efficacy.

Transplantation of corneal epithelial stem cells by limbal allograft restored useful vision to some patients with severe ocular surface disorders [166]. LESCs have been investigated as the main cell therapeutic candidates for corneal disorders, with clinical and/or preclinical studies demonstrating that LESCs cultured on an amniotic membrane effectively inhibited inflammation and reconstruct the injured corneal surface [167]. However, although LESCs have been widely used in clinical practice, visual recovery after successful transplantation remains sub-optimal [168]. MSCs were used for corneal repair include BMSCs [45], ADSCs [169], and UCB-MSCs [170]. Allogeneic MSCs exhibited good immunosuppressive effects in a pre-sensitized rat model of corneal transplantation [171]; however, no comparative studies are evaluating the effects of different MSC types on cornea reconstruction.

MSC transplantation successfully reconstructed the damaged corneal surface in a rat model, and the main therapeutic effect of MSCs was not epithelial differentiation but instead an inhibition of inflammation and angiogenesis following transplantation [172]. Transplanted amniotic fluid-derived MSCs reduce neovascularization and promote an anti-inflammatory and anti-fibrotic environment [167]. Several studies have investigated the differentiation processes associated with corneal reconstruction. Rabbit BMSCs promoted the healing of injured corneal epithelium and could be induced to differentiate into corneal epithelial-like cells expressing CK3, a corneal epithelium-specific marker, in vivo and ex vivo [173], and differentiated MSCs cultured on an amniotic membrane expressed the CK3 marker [46]. MSCs cultured in the keratocyte-conditioned medium differentiated into keratocyte-like cells [169], and Yamashita et al. [170] induced UCB-MSC differentiation into tissue-engineered corneal endothelial cells in vitro, which subsequently displayed improved preservation of corneal thickness and transparency. Further investigation is necessary to reveal whether MSCs play a role in corneal reconstruction through their transdifferentiation effects.

The immunoregulatory, anti-inflammatory, and immunosuppressive capacities of MSCs play essential roles in their therapeutic effects and have been demonstrated in studies of their transdifferentiation effects [172,174]. The associated mechanisms include downregulation of CD45 and IL-2 expression [172], downregulation of macrophage inflammatory protein-1α and VEGF [175], secretion of soluble factors [176], suppression of T cell proliferation [177], reduction of activation and migration of chemokine receptor-7-antigen-presenting cells [174], and abortion of early inflammatory responses by secretion of tumor necrosis factor-α-stimulated gene/protein-6 (TSG-6) [178]. MSCs employed as therapeutic agents for reconstructing corneas in animal models have resulted in mostly good prognoses [46].

Most available studies describe the strategies of MSC-based corneal reconstruction as involving their anti-inflammatory effects associated with corneal angiogenesis and their ability to differentiate into keratocyte-like cells; however, their differentiation efficiency and purity need to be improved.

### 4.9. Tracheal Reconstruction

Tracheal stenosis and other severe diseases, such as tracheobronchial cancer, require partial tracheal resection and reconstruction. Recently, clinical trials have been undertaken to promote tracheal reconstruction using tracheal substitutes, including autologous grafts, homografts, and prostheses [179]. Based on the progress in MSC-based tissue regeneration, investigations have been conducted concerning their efficacy for tracheal reconstruction.

For regeneration of airway epithelium, different culture methods, such as co-culture with epithelial cells or in combination with inducing factors, such as VEGF, brain-derived neurotrophic factor, TGF-β1, and activin A, have been explored. These four factors play a vital role in MSC differentiation into epithelial cells by triggering the appropriate signaling pathways [180]. Other induction factors also promote epithelial-lineage differentiation, including epidermal growth factor (EGF), keratinocyte growth factor, HGF, and IGF-2 [181]. Physical culture conditions are also involved in regulating epithelial-lineage differentiation, with studies revealing that compartmentalized or polarized culture conditions provide a suitable environment for MSC differentiation into epithelial progenitor cells with tight junction formation [182]. Similarly, BMSCs and porous tracheal scaffolds implanted in vivo after co-culture in vitro maintain their structural integrity and significantly reduce immune rejection [183].

For whole trachea regeneration, preclinical trials using animal models verified the efficacy and safety of MSC-based tracheal reconstruction. Seeding MSCs onto decellularized trachea scaffolds represents a promising means of trachea engineering in rats [184]; an acellular tracheal matrix inoculated with BMSCs showed excellent biocompatibility and immunogenicity [185]. A 3D-bioprinted artificial scaffold coated with MSCs seeded in fibrin was constructed to repair partial tracheal defects, resulting in successful restoration in the absence of graft rejection in four rabbits [186]. An aortic allograft for trachea replacement with MSC transplantation was also demonstrated as a feasible reconstruction method for tracheal defects [187]; Jorge et al. [188] used an acellular amniotic membrane for tracheal reconstruction, revealing that the membrane promoted cartilage regeneration, neovascularization, and epithelialization and reduced the risk of postoperative complications, such as tracheal stenosis. However, low porosity of the acellular tracheal matrix might lead to incomplete cartilage regeneration [189]; therefore, they used laser micropore technology to alter scaffold porosity, with in vivo results confirming that the technique improved the cartilage matrix and the mechanical strength of the scaffolds. Furthermore, artificial biomaterial-based scaffolds seeded with MSCs, such as Col-based electrospun scaffolds [190] and core−shell nanofibrous scaffolds [191], represent viable options for replacing a damaged trachea, although scaffolds loaded with MSCs promote epithelium formation and angiogenesis in vivo but in the absence of cartilage formation [192].

A tissue regeneration trial involving tracheal reconstruction supported the use of MSCs for clinical tracheal tissue engineering based on their contribution to grafted tissue integration and angiogenesis [193]. In 2008, the first transplantation of a tissue-engineered trachea was performed on a 30-year-old woman with end-stage left-main bronchus malacia and involved a trachea comprising decellularized donor trachea, epithelial cells, and BMSC-derived chondrocytes [194]. At a five-year follow-up, the patient exhibited normal lung function, ciliary function, cough reflex, and mucus clearance in the absence of stem cell-related teratoma and anti-donor antibodies [195]. This result demonstrated that the cell-seeded scaffold was clinically safe and feasible, and was followed by another BMSC-based tissue-engineered tracheal replacement in a 12-year-old boy, which also showed good results at a two-year follow-up [196].

To reconstruct a functional trachea, collaborative techniques, including those associated with stem cell biology, biomaterials, and tissue engineering, are essential, with the trachea required to assemble around the differentiating MSCs and matrix components.

### 4.10. Skin Regeneration

Skin is the first line of defense against microorganisms or physical damage, with infections and/or trauma potentially leading to skin defects. Treatment of large-scale skin damage is sometimes inadequate because autologous transplantation of skin is limited by skin tissue availability. Additionally, allogeneic skin grafts always cause immunological rejection or communicable illnesses.

Wound healing is a complex process involving interactions between soluble mediators, the ECM, and infiltrating blood cells [197]. MSC-based therapy combined with artificial scaffolds offers a promising strategy to promote wound healing or complete reconstruction of full-thickness skin (Figure 5) [198]. Feldman et al. [199] used TGF-β3, albumin-based scaffolds, and MSCs to treat pressure ulcers, with good therapeutic results. Recent progress has also been reported in skin regeneration via MSC-based therapy; MSCs significantly improve wound condition and angiogenesis. MSC-secreted TSG-6 improves wound healing by limiting macrophage activation, inflammation, and fibrosis [200]; VEGF secreted by MSCs promotes keratinocyte-mediated wound healing [201]. Angiotensin II promotes BMSC differentiation into keratinocytes through the mitogen-activated protein kinase/JNK/Janus kinase 2 signaling pathways [202]. Furthermore, gene-modified MSCs provide another possible way to promote skin regeneration. EGF-transfected MSCs [203], adenovirus-transfected C-X-C motif chemokine receptor-4-overexpressing BMSCs [204], VEGF-modified human UCB-MSCs [205], stromal cell-derived factor-1-transfected BMSCs [206], and ectodysplasin-modified MSCs [207] promote MSC-mediated wound healing activity in skin defects, as does physical stimuli, such as laser therapy [208]. A large number of scaffolds, such as those involving fibrin hydrogels [209], 3D-hybridized chitosan (CS) and poly(ε-caprolactone) (PCL) [210], Col−CS [211], sodium carboxymethylcellulose [212], and electrospun nanofibrous silk fibroin [213], have been developed to support MSC-based regeneration of defective skin. Qi et al. [214] developed a photo-cross-linkable sericin hydrogel to repair full-thickness skin damage; this hydrogen inhibited inflammation, stimulate angiogenesis, and recruit MSCs to the site of injury to regenerate skin appendages.

Recently autologous and allogeneic MSCs were both transplanted into humans to promote the regeneration of skin defects, and several clinical trials have been conducted involving the efficacy of MSC transplantation alone or combined with grafts for treating severe skin burns [215], perianal fistula [216], non-healing ulcers due to diabetes [217], dystrophic epidermolysis bullosa [218], and radiation-related skin lesions [219]. Estrogen treatment repairs diabetic wounds by significantly increasing MSC viability and proliferation and promoting neovascularization [220]. For appendage regeneration, Sheng et al. [221] reported the successful transplantation of MSCs to regenerate functional sweat glands in five patients in 2009, followed by subsequent reports of this procedure was successfully performed, with ectodysplasin-modified MSCs [207] and BMSCs on EGF-loaded scaffolds [222] demonstrated to differentiate into the functional sweat-gland cells.

### 4.11. Other Examples of MSC-Based Therapeutics

Systemic administration of MSCs alters the course of kidney injury via paracrine and/or endocrine mechanisms [223]. Intratracheal injection of BMSCs was demonstrated as a feasible strategy for managing lung diseases [30]; another study described the promotion of bladder regeneration by MSCs seeded onto PCL/CS scaffolds [224]. MSCs loaded into bioprinted vascularized tissue resulted in thicker tissue than that produced using current bioprinting methods and capable of surviving for only short periods [225]. Current evidence supports the use of MSCs for tissue regeneration in a variety of scenarios.

## 5. Potential Risk of Implanting MSCs

Although a large number of preclinical and clinical studies have been reported, the safety of MSC-related therapies remains the biggest problem for clinical applications. The principal risks of MSCs are tumorigenicity, proinflammation, and fibrosis.

Tumorigenicity is one of the most severe risk factors. On the one hand, MSCs have the ability to develop into tumors, and some studies have shown that Ewing’s sarcoma cells are derived from MSCs [226]. Additionally, a case of gliomas has occurred four years after stem cell transplantation for ataxia-telangiectasia, and the tumor cells were shown to be derived from grafts [227]. On the other hand, MSCs promote the development of tumors. Excess cytokines produced by MSCs, such as chemokines and growth factors, directly act on receptors on the surface of cancer cells, thereby regulating tumor growth. The immunosuppression ability of MSCs contributes to tumor growth and tumor cell metastasis [228,229]. Additionally, MSCs have pro-angiogenic functions in the context of tumor development [230].

MSCs show immunosuppressive effects when exposed to sufficiently high levels of pro-inflammatory cytokines. However, they promote inflammatory responses in the presence of low levels of TNF-α and IFN-γ [231]. It demonstrates that MSCs need to be triggered by inflammatory cytokines to become immunosuppressants, and the inflammatory environment is a crucial factor affecting the immune regulation of MSCs.

In addition to tissue repair, MSCs also produce fibrotic reactions. For instance, MSCs can differentiate into myofibroblasts [232]. Additionally, the balance between repair and fibrosis of MSCs is broken in the process of injury repair, which leads to fibrotic lung disease [233].

In order to improve the therapeutic effect of MSCs and reduce the potential risks, some measures should be implemented, such as reducing excessive cytokines, further exploring the immunomodulatory effects of MSCs, and establishing strict preclinical biosafety testing rules. Fortunately, major adverse events were rare according to clinical trials evaluation of MSC therapy [234]. However, this study only proves that MSCs are well tolerated and safe in the short term. As the development of cancer is a continuous process, further longer and larger controlled clinical trials are still necessary to determine the safety of MSCs.

## 6. Conclusions and Perspectives

The magic capability of regeneration of damaged parts of the body to regain lost function has long been a dream of humanity. It has been 50 years since MSCs were first identified, and advancements in the MSC-based tissue engineering have followed. In recent years, optimizations of extraction, culture, and differentiation methods have allowed MSCs to progress closer to clinical applications for disease therapy and tissue reconstruction. Three MSC properties make them optimal for tissue regeneration: (1) Immunoregulatory capacity beneficial to alleviate abnormal immune responses, (2) paracrine or autocrine functions that generate growth factors, and (3) the ability to differentiate into target cells. Previous studies of MSC-based regenerative medicine mainly focused on musculoskeletal tissues; however, recent progress has expanded their applications into other tissues, including the CNS, heart, liver, cornea, and trachea. Induction factors are one of the most critical factors affecting the outcome of MSC therapy, which sharply accelerate the repair process of MSCs on tissues. Scaffolds provide the environment for proliferation and differentiation of MSCs, and produce some mechanical stimulation to MSCs, which is beneficial for further applications of MSCs. Moreover, scaffolds loaded with induction factors enhance the therapeutic effects of MSCs, which is also worthy of further study. Scaffolds and induction factors remain indispensable agents in these processes; therefore, future investigation of advanced materials and efficient inducing factors will promote the further applications of MSCs in regenerative medicine.

Although MSCs have several advantages, there are still many challenges to overcome. The unique immunomodulatory properties of MSCs are essential for their functions, but the mechanisms of MSC immune regulation have not been elucidated. Different researchers have their distinct methods of isolating and culturing MSC, although the primary medium is similar. However, different culture conditions, such as FBS, supplements, cell seeding density, and oxygen, may affect cell proliferation and differentiation potential [235,236,237]. Therefore, a standard protocol needs to be formulated for the in vitro culture of MSCs. Additionally, cryopreserved MSCs have low viability, which will affect the further applications of cells [238]. Moreover, the age of the donors also affects the proliferation and differentiation potential of MSCs, and MSCs from young donors show lower damage and better proliferation [239]. We conclude that many factors influence the therapeutic potential of MSCs, such as induction factors, oxygen concentrations, and mechanical stimuli. Therefore, optimizing the culture conditions of MSCs may be an effective means to improve the therapeutic potential of MSCs to achieve tissue repair successfully.

For clinical applications, autologous and allogeneic MSCs have both been reported as sources for tissue regeneration. Specifically, autologous MSCs represent the primary sources considered safe for transplantation and minimization of immunological risk, despite the lack of documented complaints regarding allogeneic MSC-based therapy. The effect of MSCs on human clinical outcomes has not readily achieved the predominantly positive outcomes in murine. Furthermore, the oncogenic potential of uncontrolled MSC differentiation needs to be further investigated. The differentiation potential, surface markers, and transcription of various tissue-derived MSCs are challenging to be unified, which undoubtedly become a hindrance to the clinical transformation of MSCs.

In the clinic, the effect of MSCs on human clinical outcomes has not readily achieved the predominantly positive outcome in murine. The different results mainly attribute to the immunocompatibility and fitness of MSCs. Expanding the indications of diseases and reducing the differences among different individuals are challenges in the future research of mesenchymal stem cells. Further research is required on cell physiology about how MSCs function in vivo and how to achieve accurate administration.

Despite the current challenges, MSC-based tissue engineering represents a promising clinical strategy in the field of regenerative medicine. Moreover, improving the cultural environment of MSCs and selecting appropriate scaffolds and induction factors are essential for MSC therapy.

## Figures and Tables

**Figure 1 cells-08-00886-f001:**
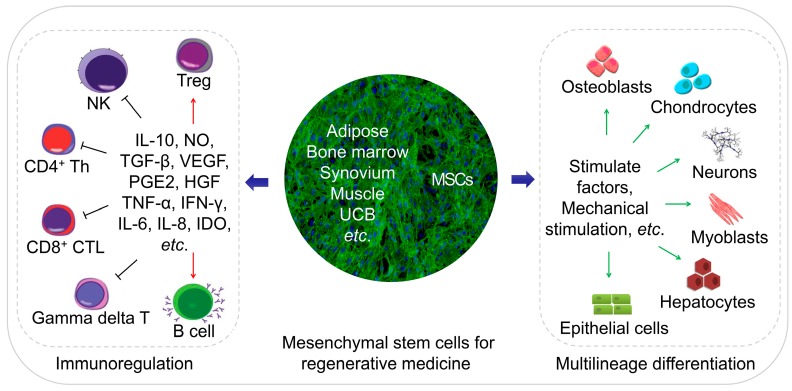
Schematic diagram of regenerative medicine based on mesenchymal stem cells (MSCs). The MSCs can be easily extracted from varies tissues, and the multilineage differentiation and immunoregulatory properties of MSCs make them an ideal cell therapeutic candidate.

**Figure 2 cells-08-00886-f002:**
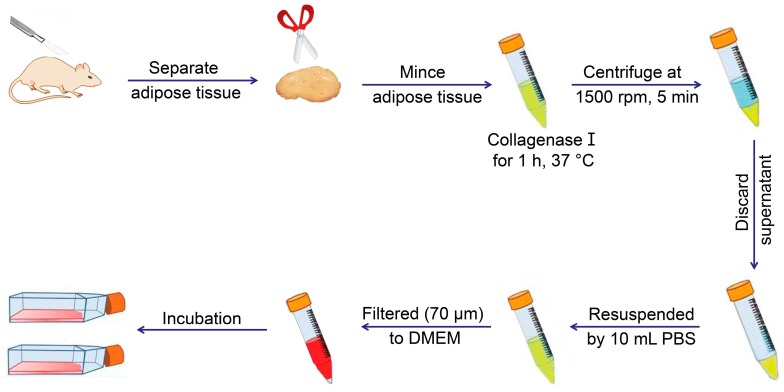
Typical extraction process of adipose-derived mesenchymal stem cells from adipose tissue of mouse.

**Figure 3 cells-08-00886-f003:**
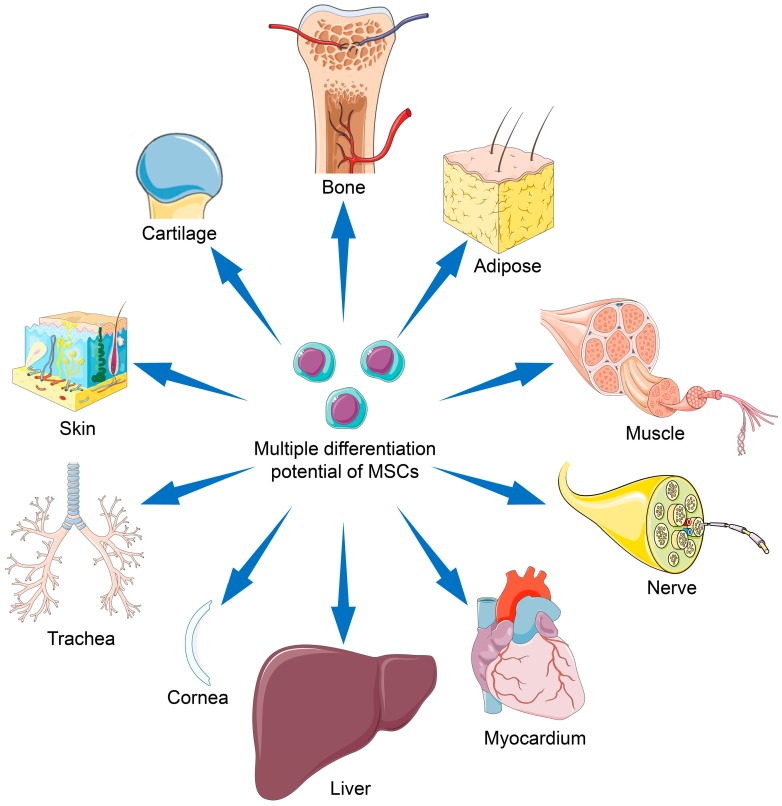
Applications of mesenchymal stem cells with multiple differentiation potential for repair of various tissues.

**Figure 4 cells-08-00886-f004:**
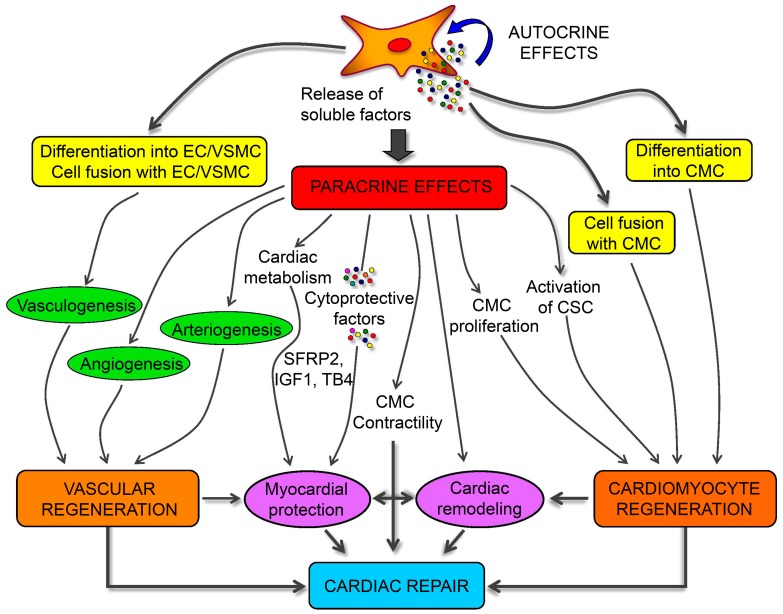
Schematic mechanisms of mesenchymal stem cells (MSCs) for cardiac regeneration. Angiogenesis, vasculogenesis, and cardiomyocytes differentiation capacities of MSCs make them possible for cardiac repair. Moreover, the paracrine effects of MSCs provide different kinds of growth and anti-inflammatory factors for the immunoregulation after ischemia of heart [139].

**Figure 5 cells-08-00886-f005:**
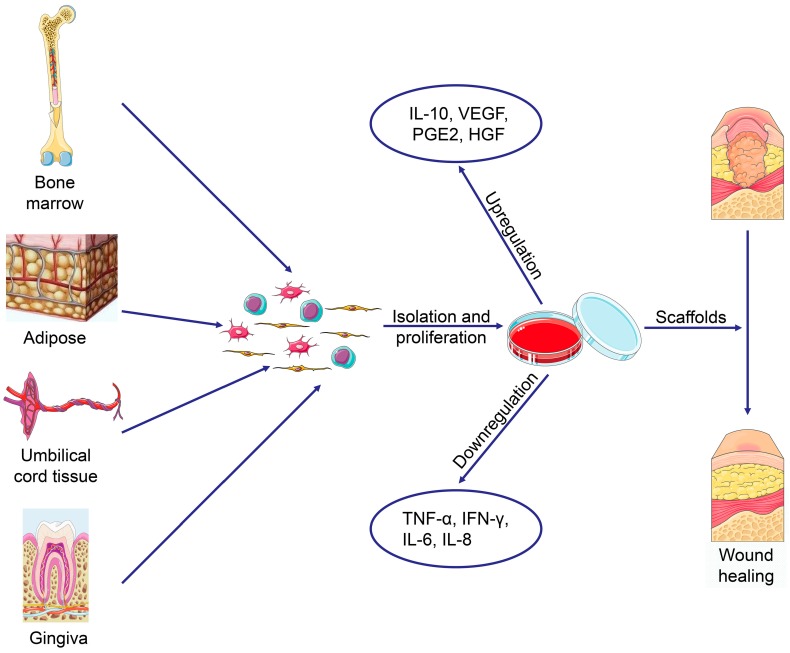
Strategies for wound healing: various mesenchymal stem cells (MSCs) are isolated and identified, and then the cells are augmented and differentiated in a specific culture condition. To realize the skin regeneration, MSCs secrete numerous factors to modulate inflammation and induce angiogenesis.

**Table 1 cells-08-00886-t001:** Extraction, discrimination, and culture of MSCs derived from various tissues.

MSC Type	Source	Extraction Approach	Culture Medium	Marker	Reference
BMSCs	Human: tubular bones and iliac crest bone marrow	1. Aspirate 1 mL of bone marrow for bone canal; 2. Extraction is diluted in PBS (1:1) and centrifuged for 30 min at 3000 rpm; 3. The obtained buffy coat is isolated, washed, and plated on culture flasks for incubation	LG-DMEM with 1% (*W*/*V*) antibiotic/antimycotic, 10% (*V*/*V*) FBS	CD29^+^, CD44^+^, CD73^+^, CD90^+^, CD105^+^, Sca-1^+^, CD14^−^, CD34^−^, CD45^−^, CD19^−^, CD11b^−^, CD31^−^, CD86^−^, Ia^−^, and HLA-DR^−^	[13,14,15]
	Mouse, rat, and rabbit: tubular bones, e.g., femurs and tibias	1. Collect femurs and tibias, cleanse the tissue with scissors, and wash the bones with 70% (*V*/*V*) ethanol and then PBS; 2. Cut off the proximal and distal parts of bones, and flush out bone marrow from bone canal by a spring to culture flasks for incubation; 3. At days 3–5, non-adherent cells are removed		Mouse: CD29^+^, CD44^+^, CD73^+^, CD90^+^, CD105^+^, Sca-1^+^, CD14^−^, CD34^−^, CD45^−^, CD11b^−^, CD31^−^, Vcam-1^−^, C-Kit^−^, CD135^−^, CD11b^−^, Ia^−^, and CD86^−^	[12,13,14,16]
				Rat: CD29^+^, CD44^+^, CD54^+^, CD73^+^, CD90^+^, CD105^+^, CD106^+^, Sca-1^+^, CD14^−^, CD34^−^, CD45^−^, and CD11b^−^	[17,31]
				Rabbit: CD29^+^, CD44^+^, CD73^+^, CD81^+^, CD90^+^, CD166^+^, CD14^−^, CD34^−^, CD45^−^, CD117^−^, and HLD-DR^−^	[15]
ADSCs	Human: subcutaneous adipose in abdomen, buttocks, and abdominal zone	1. Separate adipose from host body, and mince it with scissors or scalpel; 2. Digested by collagenase type I for 1 h at 37 °C gently shaking in a water bath; 3. Centrifuge the sample and discard the superior lipid layer; 4. Filtered through 100 and 40, or 70 μm filters; 5. Washed by 10 mL PBS and centrifuged again; 6. Discard the supernatant, resuspend the cells, and transfer them to culture flask for incubation	DMEM with 1% (*W*/*V*) P/S, 10% (*V*/*V*) FBS	Human: CD29^+^, CD44^+^, CD73^+^, CD90^+^, CD105^+^, CD146^+^, CD166^+^, MHC-I^+^, CD31^−^, CD45^−^, and HLA-DR^−^	[18,19,20,21]
	Mouse, rat, and rabbit: subcutaneous adipose			Mouse: CD34^+^, CD44^+^, CD45^+^, CD90^+^, MHC-I^+^, MHC-II^+^, and CD117^−^.	[30]
				Rat: CD44^+^, CD73^+^, CD90^+^, MHC-I^+^, CD31^−^, and CD45^−^	
				Rabbit: CD44^+^, CD105^+^, NG2^+^, CD34^−^, and CD45^−^	[22,23]
SMSCs	Synovium, especially in knee joints, of human, mouse, rat, rabbit, pig, etc.	1. Separate synovium from host knee joint, and mince it with scissors or scalpel; 2. Digested by collagenase type II, D or P at 37 °C, and filtered through a 70 μm nylon filter; 3. The released cells are washed and resuspended in a culture medium for incubation	DMEM or αMEM with 1% (*W*/*V*) P/S, 250 ng mL^−1^ amphotericin B, and 10% (*V*/*V*) FBS	Human: CD10^+^, CD13^+^, CD49^+^, CD44^+^, CD73^+^, CD90^+^, CD105^+^, CD147^+^, CD166^+^, CD14^−^, CD20^−^, CD31^−^, CD34^−^, CD45^−^, CD62^−^, CD68^−^, CD113^−^, CD117^−^, HLA-DR^−^, and ALP^−^	[24]
				Mouse: CD29^+^, CD44^+^, CD90^+^, CD34^−^, CD45^−^, and CD107^−^	[25]
				Rat: CD90^+^, CD11b^−^, and CD45^−^	[31]
				Rabbit: CD44^+^, CD90^+^, and CD105^+^	[26,27]
UCB-MSCs	Umbilical cord blood of human	1. Harvest of human umbilical cord blood; 2. Mononuclear cells (MNC) are isolated from the buffy coat layer; 3. Seed into 25 cm^2^ flask, and non-adherent cells are removed after 48 h	LG-DMEM, 1% P/S, 250 ng mL^−1^ amphotericin B, and 10% (*V*/*V*) FBS	CD29^+^, CD44^+^, CD73^+^, CD90^+^, CD105^+^, CD166^+^, CD14^−^, CD31^−^, CD34^−^, CD45^−^, CD106^−^, and HLA-DR^−^	[28]

**Table 2 cells-08-00886-t002:** Summary of differentiation researches and application potential of MSCs.

Differentiation Direction *	Preferred MSC Type	Basic Induction Medium		Identify Methods				Application Field	Reference
		Basic Medium	Induce Agents	Staining	IHC	RT-PCR	Others		
Osteoblast	BMSCs	LG-DMEM, 10% (*V*/*V*) FBS, 1% (*W*/*V*) antibiotic/antimycotic (In some studies, the osteogenic medium used HG-DMEM solution)	10.0 mM β-glycerophosphate, 50.0 μg mL^−1^ ascorbic acid, and 100 nM dexamethasone	**Alizarin red staining**, Von Kossa Staining	**Col I, OCN, OPN**	**Col I, OCN, OPN, ALP**, BSP, Osterix, RUNX2	**ALP activity**, Calcium assay kit	Bone regeneration	[32]
Chondrocyte			50.0 μM ascorbic acid, 100 nM dexamethasone, 10.0 ng mL^−1^ TGF-β1/TGF-β3	**Alcian blue staining, Toluidine blue staining**	**Col II**	**Col II, SOX-9, Aggrecan**, SOX-5, SOX-6, NOX 4, Col X, Chondroitin 4-sulfotransferase	GAG assay kit	Cartilage regeneration	[33,34]
Neurocyte	BMSCs, ADSCs		10.0 ng mL^−1^ EGF, 20 ng mL^−1^ HGF, 20 ng mL^−1^ VEGF; 8 days later, 200 µM BHA, 5.0 mM KCl, 2.0 µM valproic acid, 10 µM forskolin, 1.0 µM hydrocortisone, and 5.0 µM insulin are added to the medium	—	**Enolase**, **Tubulin-βIII**, **GFAP**, S100, MBP, MAP2, NF	**Tubulin-βIII**, **GFAP**, **Enolase**, NeuN, NCAM, Glial cell marker, NANOG, OCT4 and SOX-2, MAP2, NF-M, GAP 43	—	Nerve regeneration	[35,36,37]
Cardiomyocyte	ADSCs		10.0 µg L^−1^ bFGF, 10.0 µM 5-azacytidine; one day later, the medium maintained in the same conditions without 5-azacytidine for four weeks	—	Desmin, M-cadherin, MHC, α-cardiac actin, cTnI	Desmin, MYOD1, MYOG, MHC, α-cardiac actin, cTnT, MYF5/6, MEF2C, TNNI1/2, CKM, Myosin2, HCN2, HCN4	Heterotypic Cell Fusion Assay	Myocardial regeneration	[38,39,40]
Hepatocyte	PDSCs		1X ITS, 10^−8^ M dexamethasone, 20.0 ng mL^−1^ EGF, 20.0 ng mL^−1^ FGF, 40.0 ng mL^−1^ OsM, 40 ng mL^−1^ HGF; After two weeks, the medium is replaced with hepatic differentiation medium with an increased concentration of dexamethasone at 10^−5^ M and/or 1.0 µM TSA	**PAS staining**	ALB, AFP, CK-18, PanCK, CK 19, Transthyretin	ALB, **AFP**, β-actin, CK-18, HNF-4α, Transthyretin, TDO2, and CYP7A1	LDL/CM-Dil uptake assay; Cell morphology; Ammonia clearance; Albumin production; ELISA assay	Liver regeneration	[41,42,43,44]
Keratocyte	No comparative studies	LG-DMEM: F-12 3:1, 5% FBS, 1% (*W*/*V*) antibiotic/antimycotic	Induction medium: without pyruvate, 25.0 ng mL^−1^ BMP-4, 1.0 mM all-trans retinoic acid, and 10.0 ng mL^−1^ EGF; Differentiation medium: 5.0 µg mL^−1^ insulin, 2.0 nM tri-iodothyronine, 2.0 nM adenine, and 10.0 ng mL^−1^ EGF	**H&E staining**	CK3, β1-integrin, and E-cadherin, p63, CK12, CK8, CK14, CK15	ABCG2, β1-integrin, CEBPδ, CK3, and p63, Oct4, Sox2, Nanog, Rex1, DSC1, and DSG1	Transepithelial electrical resistance	Corneal regeneration	[45,46]

* The culture media for another end-stage lineage cells have not been standardized except osteoblasts and chondrocytes. The bold words in identified methods mean the main identified staining, proteins of immunohistochemistry, genes of real-time reverse transcription polymerase chain reaction (RT-PCR), and other methods.

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
