# Peer review of "Mesenchymal Stem Cells for Regenerative Medicine"

_cells, 2019, doi:10.3390/cells8080886_

Round 1

Reviewer 1 Report

The review on mesenchymal stem cells (MSCs) for regenerative medicines by Han et al. mainly describes the use of MSCs to regenerate organs and tissues. Unfortunately, the authors do not address the potential risk of implanting MSCs, including the safety issue, profibrogenic risk, tumor risk or even the long-term safety with MSCs. This is my major concern. I found that an entire section about these risks should be included in the review. 

The achievements with MSCs on some organs like the central nervous system and the differentiation methods should be emphasised by adding a scheme or figure similarly to the figure 3. It helps summarising the current knowledge.

These improvements should be added in a minor revision.

Author Response

Response to Reviewer 1:

The review on mesenchymal stem cells (MSCs) for regenerative medicines by Han et al. mainly describes the use of MSCs to regenerate organs and tissues. Unfortunately, the authors do not address the potential risk of implanting MSCs, including the safety issue, profibrogenic risk, tumor risk or even the long-term safety with MSCs. This is my major concern. I found that an entire section about these risks should be included in the review.

The achievements with MSCs on some organs like the central nervous system and the differentiation methods should be emphasized by adding a scheme or figure similarly to the Figure 3. It helps summarizing the current knowledge.

These improvements should be added in a minor revision.

Response: Thanks for your comments and constructive suggestions. In the revised manuscript, we have added a new section on the potential risks of MSCs implantation and briefly discussed MSC safety issues such as tumor risk, proinflammatory risk, and profibrotic risk. Besides, we have added a paragraph with a new figure in section 4 (MSC-Based Regenerative Medicine) to illustrate the application of MSCs in tissue repair, and we also added some text to improve the logic of the manuscript.

The Potential Risk of Implanting MSCs

Although a large number of preclinical and clinical studies have been reported, the safety of MSCs-related therapies remains the biggest problem for clinical applications. The main risks of MSCs are the tumorigenicity, proinflammation, and fibrosis.

Tumorigenicity is one of the most severe risk factors. On the one hand, MSCs have the ability to form tumors, and some studies have shown that Ewing's sarcoma cells are derived from MSCs [226]. Additionally, a case of gliomas has occurred 4 years after stem cell transplantation for ataxia telangiectasia, and the tumor cells were shown to be derived from grafts [227]. On the other hand, MSCs promote the development of tumors. Excess cytokines produced by MSCs, such as chemokines and growth factors, directly act on receptors on the surface of cancer cells, thereby regulating tumor growth. The immunosuppression ability of MSCs contributes to tumor growth and tumor cell metastasis [228, 229]. Additionally, MSCs have pro-angiogenic functions in the context of tumor development [230].

MSCs show immunosuppressive effects when exposed to sufficiently high levels of pro-inflammatory cytokines, however, they promote inflammatory responses in the presence of low levels of TNF-α and IFN-γ [231]. It demonstrates that MSCs need to be triggered by inflammatory cytokines to become immunosuppressants, and the inflammatory environment is a key factor affecting the immune regulation of MSCs.

In addition to tissue repair, MSCs also produce fibrotic reactions. For instance, MSCs have the ability to differentiate into myofibroblasts [232]. Additionally, the balance between repair and fibrosis of MSCs is broken in the process of injury repair, which can lead to fibrotic lung disease [233].

In order to improve the therapeutic effect of MSCs and reduce the potential risks, some measures should be implemented, such as reducing excessive cytokines, further exploring the immunomodulatory effects of MSCs, and establishing strict preclinical biosafety testing rules. Fortunately, major adverse events were rare according to clinical trials evaluation of MSCs therapy [234]. However, this study only proves that MSCs are well tolerated and safe in the short term. As the development of cancer is a chronic process, further longer and larger controlled clinical trials are still necessary to determine the safety of MSCs.

Figure 3. Applications of MSCs with multiple differentiation potential for repair of various tissues.

MSC-Based Regenerative Medicine

So far, MSCs have been widely studied and applied in regenerative medicine. In this section, we summarize reports concerning the latest preclinical and clinical trials of various MSC types for tissue engineering. The topics mainly focus on reconstruction of fragile tissues, including those associated with the musculoskeletal system, nervous system, myocardium, liver, cornea, trachea, and skin, as shown in Figure 3.

Discovery and Extraction of MSCs from Different Sources

The rich source of MSCs is the critical basis for their extensive researches and applications. It is known that MSCs can be isolated from various tissues, such as bone marrow, adipose, and synovium, and human umbilical cord blood, and bone marrow is one of the most essential sources of MSCs.

Differentiation Potentials of MSC Types

Multi-directional differentiation potential is one of the most important characteristics of MSCs. In addition, different tissue sources affect the differentiation tendency and proliferation capability of MSCs.

4.2. Cartilage Repair

Cartilage defect repair is one of the major challenges faced by orthopedic surgeons. Due to the inherent avascular nature of cartilage and the proliferation of mature chondrocytes, cartilage is greatly limited in its ability to repair itself. Currently clinically applied cartilage repair techniques, such as bone marrow stimulation and osteochondral transplantation, have their own limitations. Fibrocartilage produced by bone marrow stimulation is not strong enough, and grafts for osteochondral transplantation are difficult to integrate.

4.3. Regeneration of Other Musculoskeletal Tissue

Recent studies investigated the MSC-based tissue regeneration of musculoskeletal tissue outside of bone and cartilage, including the meniscus, tendons and ligaments, and intervertebral discs (IVD).

4.4. Central Nervous System Rebuilding

Adult central nervous system (CNS) lacks the ability to repair damaged neurons, so the damage of CNS is irreversible, and there is currently no effective repair method for CNS injury in clinical practice repair.

4.6. Myocardium Restoration

Cardiac disease is characterized by substantial morbidity and mortality, and serious adverse consequences. In addition to congenital heart disease, almost all cardiac diseases involve insufficient blood supply to key regions, resulting in myocardial damage and necrosis. Although myocardium has limited regenerative capacity, restoration of severe damage to cardiomyocytes due to catastrophic myocardial infarction or other myocardial diseases is inadequate.

4.7. Liver Regeneration

The liver is an important human organ, failure of which can cause fatal illnesses.

4.8. Corneal Reconstruction

The cornea represents a transparent avascular connective tissue that provides most of the refractive ability of the eye and acts as the primary barrier against infection and mechanical damage to internal structures. Because the cornea is fragile and directly exposed to the external environment, a variety of clinical disorders, such as aniridia and Stevens–Johnson syndrome, or chemical, mechanical, and thermal injury can potentially cause corneal injury.

4.10. Skin Regeneration

Skin is the first line of defense against microorganisms or physical damage, with infections and/or trauma potentially leading to skin defects. Treatment of large-scale skin damage is sometimes inadequate, because autologous transplantation of skin is limited by skin-tissue availability. Additionally, allogeneic skin grafts can cause immunological rejection or communicable illnesses.

Reviewer 2 Report

The authors present in this work a summary of extraction methods and subsequent potential for differentiation of mesenchymal stem cells, and discuss further some of their preclinical and clinical applications in regenerative medicine. Recent years revealed a strong interest the multipotent mesenchymal stromal cells, especially for the therapeutic applications, as revealed by almost 1000 registered clinical trials using MSC. Thus, this work should to be timely and of interest. However, the overall impression of this manuscript is of a draft rather than finished work, although the second part of the manuscript (point 4. MSC-Based Regenerative Medicine) is rather well composed and described. The tables provide useful overviews of MSC isolation and differentiation methods, but the introductory main text needs massive revision to improve clarity and accuracy. Also, a longer discussion on potential side effects risks associated with MSC clinical use (for example maldifferentiation, tumor growth) would be beneficial to this this work.    

Minor comments:

-        For example, please rephrase the:

o   fig1 legend: “The MSCs can be easily extracted from varies tissues, and both multilineage differentiation and immunoregulation properties of them derived MSCs to be the ideal cell therapeutic agents”-verb missing

o   table 2 “The culture media for another end-stage lineage cells are not mature except osteoblasts and chondrocytes.”- culture media cannot be mature.

-        Please check grammar, for example line 47: “MSCs exist in various tissues

-        page 9 “Reports concerning differences among various MSC types are scarce.” – actually there are an increasing number of articles addressing the heterogeneity of the MSC, please see for ex. Andrzejewska et al., 2019 (http://dx.doi.org/ 10.1002/stem.3016).

-        Please check:

o   There are not bold words in the tables, except the table head. The bold words in identify methods mean the main identify staining, proteins of immunohistochemistry, genes of RT-PCR, and other methods.

o   Repetitions, for example: “Table 1 lists a variety of markers expressed on the MSC surface.” Is present on line 51 and line 53.

-        regarding the mandibular bone regeneration, the authors might want to include the following clinical study also Gjerde et al., doi: 10.1186/s13287-018-0951-9

Author Response

Response to Reviewer 2:

The authors present in this work a summary of extraction methods and subsequent potential for differentiation of mesenchymal stem cells, and discuss further some of their preclinical and clinical applications in regenerative medicine. Recent years revealed a strong interest the multipotent mesenchymal stromal cells, especially for the therapeutic applications, as revealed by almost 1000 registered clinical trials using MSC. Thus, this work should to be timely and of interest. However, the overall impression of this manuscript is of a draft rather than finished work, although the second part of the manuscript (point 4. MSC-Based Regenerative Medicine) is rather well composed and described. The tables provide useful overviews of MSC isolation and differentiation methods, but the introductory main text needs massive revision to improve clarity and accuracy. Also, a longer discussion on potential side effects risks associated with MSC clinical use (for example maldifferentiation, tumor growth) would be beneficial to this this work.

Response: Thanks for your kind comments and suggestion. In the second part of the revised manuscript, the introductory text was supplemented to accurately describe the extraction method of the MSCs and to clearly elicit the tables. Additionally, a new section on the potential risks of MSCs implantation has been added.

Discovery and Extraction of MSCs from Different Sources

Figure 2 and Table 1 describe the general protocols used for MSC extraction. Briefly, the process involves isolation of various tissues, digestion to obtain cells, and culture for 3 to 5 days, followed by discarding non-adherent cells and continuous culture of adherent cells to the desired passage. The basic culture medium for MSCs includes low-glucose Dulbecco's modified Eagle medium (LG-DMEM) with 1% (W/V) antibiotic/antimycotic and 10% (V/V) fetal bovine serum (FBS). Additionally, Table 1 lists a variety of markers expressed on the MSC surface. Notably, rabbit is the most frequently used animal model for experiments, involving cartilage or bone tissue regeneration, and should receive increased focus concerning MSC identification. Moreover, the surface markers of rabbit-tissue-derived MSCs require further verification.

The Potential Risk of Implanting MSCs

Although a large number of preclinical and clinical studies have been reported, the safety of MSCs-related therapies remains the biggest problem for clinical applications. The main risks of MSCs are the tumorigenicity, proinflammation, and fibrosis.

Tumorigenicity is one of the most severe risk factors. On the one hand, MSCs have the ability to form tumors, and some studies have shown that Ewing's sarcoma cells are derived from MSCs [226]. Additionally, a case of gliomas has occurred 4 years after stem cell transplantation for ataxia telangiectasia, and the tumor cells were shown to be derived from grafts [227]. On the other hand, MSCs promote the development of tumors. Excess cytokines produced by MSCs, such as chemokines and growth factors, directly act on receptors on the surface of cancer cells, thereby regulating tumor growth. The immunosuppression ability of MSCs contributes to tumor growth and tumor cell metastasis [228, 229]. Additionally, MSCs have pro-angiogenic functions in the context of tumor development [230].

MSCs show immunosuppressive effects when exposed to sufficiently high levels of pro-inflammatory cytokines, however, they promote inflammatory responses in the presence of low levels of TNF-α and IFN-γ [231]. It demonstrates that MSCs need to be triggered by inflammatory cytokines to become immunosuppressants, and the inflammatory environment is a key factor affecting the immune regulation of MSCs.

In addition to tissue repair, MSCs also produce fibrotic reactions. For instance, MSCs have the ability to differentiate into myofibroblasts [232]. Additionally, the balance between repair and fibrosis of MSCs is broken in the process of injury repair, which can lead to fibrotic lung disease [233].

In order to improve the therapeutic effect of MSCs and reduce the potential risks, some measures should be implemented, such as reducing excessive cytokines, further exploring the immunomodulatory effects of MSCs, and establishing strict preclinical biosafety testing rules. Fortunately, major adverse events were rare according to clinical trials evaluation of MSCs therapy [234]. However, this study only proves that MSCs are well tolerated and safe in the short term. As the development of cancer is a chronic process, further longer and larger controlled clinical trials are still necessary to determine the safety of MSCs.

Minor comments:

- For example, please rephrase the:

o Fig1 legend: “The MSCs can be easily extracted from varies tissues, and both multilineage differentiation and immunoregulation properties of them derived MSCs to be the ideal cell therapeutic agents”-verb missing

o Table 2 “The culture media for another end-stage lineage cells are not mature except osteoblasts and chondrocytes.”- culture media cannot be mature.

Response: Thanks for your kind advice. We have revised the relevant language issues and carefully checked and corrected the grammar of the whole manuscript.

Fig1 legend

both multilineage differentiation and immunoregulation properties of them derived MSCs to be the ideal cell therapeutic agents” has been replaced by “the multilineage differentiation and immunoregulatory properties of MSCs make it an ideal cell therapeutic agents”.

Table 2

are not mature” has been replaced by “have not been standardized”.

- Please check grammar, for example line 47: “MSCs exist in various tissues

Response: Thanks for your kind reminding. We have modified this issue.

- Page 9 “Reports concerning differences among various MSC types are scarce.” – actually there are an increasing number of articles addressing the heterogeneity of the MSC, please see for ex. Andrzejewska et al., 2019 (http://dx.doi.org/ 10.1002/stem.3016).

Response: Thanks for your suggestion. This issue has been revised as followed according to the review comments and the relevant reference has been updated.

Differentiation Potentials of MSC Types

There are an increasing number of publications addressing the heterogeneity of the MSCs [47].

References

Andrzejewska, A.; Lukomska, B.; Janowski, M., Concise Review: Mesenchymal Stem Cells: From Roots to Boost. Stem cells (Dayton, Ohio) 2019, 37, (7), 855-864.

- Please check:

o There are not bold words in the tables, except the table head. The bold words in identify methods mean the main identify staining, proteins of immunohistochemistry, genes of RT-PCR, and other methods.

o Repetitions, for example: “Table 1 lists a variety of markers expressed on the MSC surface.” Is present on line 51 and line 53.

Response: Thanks for your comments. The key identification methods in Table 2 have been changed to bold, and the error of repetition in the second part has also been corrected.

Table 2. Summary of differentiation researches and application potential of MSCs.

The bold words in identify methods mean the main identify staining, proteins of immunohistochemistry, genes of real-time reverse transcription polymerase chain reaction (RT-PCR), and other methods.

Discovery and Extraction of MSCs from Different Sources

Figure 2 and Table 1 describe the general protocols used for MSC extraction. Briefly, the process involves isolation of various tissues, digestion to obtain cells, and culture for 3 to 5 days, followed by discarding non-adherent cells and continuous culture of adherent cells to the desired passage. The basic culture medium for MSCs includes low-glucose Dulbecco's modified Eagle medium (LG-DMEM) with 1% (W/V) antibiotic/antimycotic and 10% (V/V) fetal bovine serum (FBS). Additionally, Table 1 lists a variety of markers expressed on the MSC surface. Notably, rabbit is the most frequently used animal model for experiments, involving cartilage or bone tissue regeneration, and should receive increased focus concerning MSC identification. Moreover, the surface markers of rabbit-tissue-derived MSCs require further verification.

- Regarding the mandibular bone regeneration, the authors might want to include the following clinical study also Gjerde et al., doi: 10.1186/s13287-018-0951-9

Response: Thanks for your comments and suggestions. The aforementioned excellent article has been cited in the revised manuscript as follows.

Specifically, dentists have used this technique to address alveolar cleft defects, jaw-defect reconstruction, and maxillary sinus augmentation, with excellent outcomes [66-68].

References

Gjerde, C.; Mustafa, K.; Hellem, S.; Rojewski, M.; Gjengedal, H.; Yassin, M. A.; Feng, X.; Skaale, S.; Berge, T.; Rosen, A.; Shi, X. Q.; Ahmed, A. B.; Gjertsen, B. T.; Schrezenmeier, H.; Layrolle, P., Cell therapy induced regeneration of severely atrophied mandibular bone in a clinical trial. Stem Cell Res Ther 2018, 9, (1), 213.
